# Real-Time FPGA Accelerated Stereo Matching for Temporal Statistical Pattern Projector Systems

**DOI:** 10.3390/s21196435

**Published:** 2021-09-26

**Authors:** Zan Brus, Marko Kos, Matic Erker, Iztok Kramberger

**Affiliations:** Faculty of Electrical Engineering and Computer Science, University of Maribor, 2000 Maribor, Slovenia; marko.kos@um.si (M.K.); matic.erker@um.si (M.E.); iztok.kramberger@um.si (I.K.)

**Keywords:** depth sensor, stereo vision, FPGA, HLS, temporal stereo, statistical pattern projection, hardware acceleration

## Abstract

The presented paper describes a hardware-accelerated field programmable gate array (FPGA)–based solution capable of real-time stereo matching for temporal statistical pattern projector systems. Modern 3D measurement systems have seen an increased use of temporal statistical pattern projectors as their active illumination source. The use of temporal statistical patterns in stereo vision systems includes the advantage of not requiring information about pattern characteristics, enabling a simplified projector design. Stereo-matching algorithms used in such systems rely on the locally unique temporal changes in brightness to establish a pixel correspondence between the stereo image pair. Finding the temporal correspondence between individual pixels in temporal image pairs is computationally expensive, requiring GPU-based solutions to achieve real-time calculation. By leveraging a high-level synthesis approach, matching cost simplification, and FPGA-specific design optimizations, an energy-efficient, high throughput stereo-matching solution was developed. The design is capable of calculating disparity images on a 1024 × 1024(@291 FPS) input image pair stream at 8.1 W on an embedded FPGA platform (ZC706). Several different design configurations were tested, evaluating device utilization, throughput, power consumption, and performance-per-watt. The average performance-per-watt of the FPGA solution was two times higher than in a GPU-based solution.

## 1. Introduction

Three-dimensional optical measurement systems have seen significant improvements in measurement capabilities in the past years, and this trend continues to grow. The increased technological advances in computing hardware and more advanced algorithmic approaches have also contributed to this aspect, enabling new applications in biomedical imaging, metrology, and many other areas.

With the development of three-dimensional optical metrology, different approaches to solving metrological problems have emerged. The type of approaches can generally be classified based on the presence of pattern projection units. Solutions that do not employ projection units can be described as passive approaches. The main advantages of passive approaches are the simplified hardware requirements and high measurement speeds but at the cost of reduced accuracy and precision. Solutions that utilize illumination are classified as active approaches [1]. These approaches are often more complex in hardware, due to non-trivially controlled pattern projectors emitting structured light. These projectors can be implemented in many different forms, each having its characteristic strengths and weaknesses. Popular examples of active projector-based systems can employ time-modulated light illumination (e.g., TOF), statistical pattern projection (e.g., pseudorandom dot projection), or digital/analog fringe projection, to name a few. Further information on most coded structured light projection methods can be found in [2,3].

This paper will focus on the algorithmic side of disparity calculation for a specific active projection approach called aperiodic sinusoidal fringe projection. Aperiodic sinusoidal fringe projection has become increasingly popular in systems that deliver high-speed 3D depth measurements [4,5]. Traditional binary coding methods and fringe analysis methods rely on digital pattern projection, which requires precise control of the projected pattern and a priori pattern knowledge. Depth reconstruction in such scenarios is done using codeword decoding, and in the case of controlled periodic pattern projection, using phase-measuring profilometry (PMP) or Fourier transform profilometry (FTP). In contrast, measurement systems that rely on aperiodic fringe projection methods do not require precise phase shift control of the projected fringe pattern or any prior pattern knowledge [4]. The method of disparity calculation relies on finding locally unique temporal changes in brightness to establish a pixel correspondence between at least two cameras. The minimum requirement for a working system is a projector source that provides a sufficiently large temporal variation of the projected pattern and at least two cameras observing the projected pattern. The projection of temporally varying aperiodic sinusoidal fringes represents an optimized way of aperiodic fringe projection suitable for epipolar geometries. The overall depth reconstruction quality depends on the amount of captured temporal samples [5]. The simplification of complex projector requirements simplifies the end system design, which enables designers to replace otherwise complex DLP and LCD projectors with simplified mechanical counterparts [4].

The drawback of aperiodic sinusoidal fringe projection systems is the increased depth reconstruction error for the same number of temporal measurements, compared to periodic fringe projection methods [5]. Another less commonly addressed issue is the computational burden associated with the algorithms used for disparity calculation, as highlighted in [6]. Compared to algorithms used in periodic fringe projection [1], the computational requirements of temporal pixel-to-pixel correlation limit the portability of such approaches to high-end computing devices. Utilization of computationally expensive matching cost metrics, such as normalized cross-correlation (NCC), often requires processing capabilities that are only achievable using power-demanding non-mobile GPU platforms. Addressing this computational burden on a low-power FPGA platform and the FPGA-specific design decisions at the high-level synthesis (HLS) level is the main focus of the presented paper.

Therefore, we propose a flexible HLS-based architecture for energy-efficient, real-time disparity calculation. The significant contributions of the presented work are as follows:A pipelined HLS-based architecture that utilizes a problem-specific design structure to minimize the required FPGA resources and maximize throughput.Utilization of sliding window buffer design optimizations for reduction of out-of-order memory accesses as well as balanced use of FPGA resources.Performance comparison with a GPU-based solution. We evaluate the device utilization, throughput, power consumption and energy efficiency of disparity calculation, using different configurations.

The rest of the paper is organized as follows: Section 2 provides background information and related work. Section 3 describes the architecture and optimizations; Section 4 presents the experimental results, and Section 5 provides the conclusion.

## 2. Related Work

In the past few years, several stereo-vision disparity calculation methods have been presented [7]. Most of the published research focuses on stereo systems with no active illumination—passive stereo systems. Passive stereo systems are easy to implement in hardware, as they require no active illumination means: the only hardware requirements are two synchronized camera sensors. This reduced hardware complexity makes end solutions cheaper to manufacture, more widely available, and thus, easier to obtain [2].

Systems that employ active illumination are more complex in hardware, but often offer better, more accurate results. Such systems do not necessarily require a stereo vision pair if proper and accurate knowledge about the projected pattern is available. The most used approach that uses active illumination is called sinusoidal fringe projection, utilizing phase-shift profilometry [2]. Other concepts that employ active illumination are also Time-of-Flight (ToF) as well as single and multi-shot statistical pattern projection. 

The scope of this paper is limited to aperiodic sinusoidal temporal pattern projection. Currently, some publications are evaluating the use of such methods [4,5,6,8]. However, the main subject of these publications mostly stems from the field of optics. The closest research work that addresses the issue of computational complexity in temporal statistical pattern stereo was published by Dietrich et al. [6]. The main contribution of this work was the incorporation of an extended census transform over the temporal axis and the use of the Hamming distance as the matching cost metric when calculating disparity. This simplification enabled the real-time operation of such an approach on a desktop graphical processing unit (Nvidia GTX 1080).

As noted in these publications, temporal statistical projection systems offer relaxed hardware requirements but at the cost of increased computational complexity. This computational burden becomes apparent in systems with constrained computing capabilities or in systems operating on a limited power budget. Since the domain of temporal stereo image vision has seen limited—niche—use over the years, there is still a research gap to be filled, especially regarding the computational aspect of such systems. Narrowing this gap, with emphasis on low-power design on an FPGA platform, is one of the main goals of this research paper.

## 3. Design Architecture

### 3.1. Temporal Stereo 

The goal of a stereo-vision system is the reconstruction of three-dimensional coordinates from a pair of simultaneously captured images from different viewpoints. This goal can be achieved by using a stereo camera and is commonly realized in the form of placing two cameras along a horizontal axis, separated by a baseline distance, forming an epipolar geometry [7].

After obtaining the stereo image pair, the two separate views are used to estimate the depth information. This is done by finding the best possible pixel correspondences of the same environmental features, seen from different views. The correspondence between the left and right camera view is also known as disparity [7].

Passive stereo vision systems exploit spatial coherence (e.g., texture) to establish correspondence between views. Temporal stereo vision systems try to establish the correspondence by exploiting the spatial uniqueness of temporal brightness variation of the actively illuminated scene. The correspondence is, therefore, encoded in the time domain. This additional dimension makes it possible to obtain accurate and valid correspondence between the left and right view on a pixel-level resolution if the observed scene is illuminated in a way that provides sufficient temporal variation and enough samples are available. 

An example of finding pixel-level resolution correspondence between the left and right view when illuminating the scene with an aperiodic sinusoidal fringe projector is displayed in Figure 1. 

Searching for correspondence between two separate views on a single-pixel level only works on stereo systems with active illumination. A temporal pixel can be described as the temporal (time) series of brightness values of a single pixel at constant x and y image coordinates. Accurate and valid results can only be obtained if multiple images of the scene are taken, and the changes in the illumination pattern can be observed from both views. The projected pattern needs to exhibit enough temporal variability.

### 3.2. FPGA System Architecture

In this section, we introduce the high-level structure of the proposed FPGA design. We present the design decisions that were made in order to achieve maximum throughput. All the important building blocks are presented, using snippets of HLS code. The HLS code was designed for the Vivado HLS compiler but can be easily modified to be used in any other HLS environment. The proposed structure is displayed in Figure 2.

The operation of such a design can be described, using the following high-level building blocks:AXI Input Interface—An input FIFO structure is required in order to read data from external memory and save data in a suitable order for further processing. Captured images residing in external memory (DRAM) are stored sequentially in successive memory regions, most commonly in an order determined by the time of capture. This means that accessing image pixels with the same local x and y coordinates, but taken at different time instants, requires non-sequential access. The main purpose of this structure is to buffer the respective batched number of pixels from every image in a way to maximize throughput and minimize the number of external memory accesses. Data access is done over the Advanced eXtensible Interface (AXI) interface.Temporal Census Transform—The temporal variation of brightness at a pixel level is encoded using a modified census transform operation, forming the feature vector, that can be represented as a bitstring. Grayscale pixel values (brightness) are encoded as a bitstring in the same way as that published in [6].Disparity Calculator—Using the temporally encoded features (bitstrings), both left-to-right and right-to-left disparities are calculated. The Hamming distance is used as the metric for evaluating the matching costs between bitstrings. The main goal of this unit is to obtain an initial disparity estimate, which is later refined to a subpixel level. Calculating both the left-to-right as well as the right-to-left disparities is required in order to obtain consistent results. The disparity calculation is performed, using a sliding window architecture.Consistency Checker—Consistency checking is required to ensure that the calculated disparities are valid. A calculated disparity is consistent if the difference between the left-to-right disparity and right-to-left disparity corresponding to the same point in space, is less than or equal to 1.Median Filter—Not all disparities calculated by the disparity calculator are consistent. Inconsistent disparities are present, which are marked using the consistency checker. The spatial distribution of these inconsistencies can be represented as a form of salt and pepper noise. Such inconsistencies can be mitigated by application of a median filter. A median filter can be easily realized, using the sliding window approach.Subpixel Refinement—The main goal of the disparity calculator is to obtain a coarse disparity map. This initial coarse disparity estimate can be used as a baseline for further refinement to a subpixel. Refinement is done using the normalized cross-correlation similarity measure on a reduced set of temporal pixel candidates. A sliding window approach, similar to the one used in the coarse disparity calculation, can be utilized for such a purpose.AXI Output Interface—An output FIFO structure is required in order to write the resulting subpixel disparities over the AXI interface to the external memory (DRAM).

In order to maximize the throughput, instruction and task pipelining have to be utilized. Single instruction pipelining is achieved with predictable operation and memory access scheduling inside individual building blocks. Task pipelining is achieved with proper utilization of high-level operations, but only after every high-level task is internally adequately pipelined. The high-level task schedule used in the FPGA-based design is displayed in Figure 3.

The task schedule presented in Figure 3 indicates several key points of the proposed design. The initiation interval is the time that must elapse between issuing two tasks of the same type. Because we are working with temporal data, several non-sequential read operations (non-sequential in terms of in-memory location) must be made to load the appropriate data from the external memory into the hardware accelerator. Every read operation has to be issued for a large, contiguous memory region (e.g., image line) in order to minimize the penalty of external memory access. After the data have been read from the external memory and saved in local memory structures, the computational tasks can be initiated. When the computation is finalized, the subpixel level disparity is written back to external memory.

Describing such a pipelined high-level design in the form of HLS pseudocode can be done in the way as presented in Figure 13.

**Listing 1 sensors-21-06435-f013:**
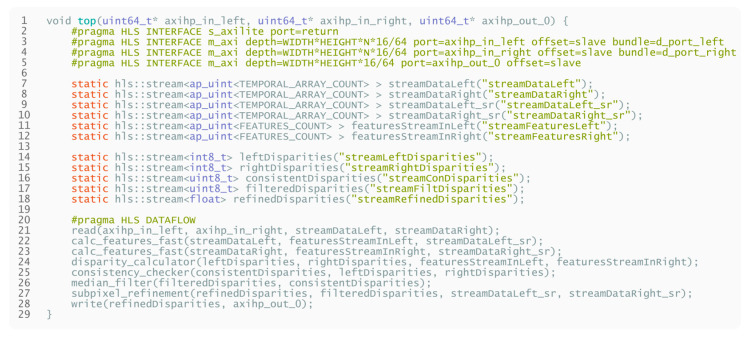
High-level pseudocode for the proposed FPGA design.

### 3.3. Feature Calculation—Temporal Census Transform

The temporal census transform procedure used in this design was previously proposed as the BICOS Plus algorithm by Dietrich et al. [6]. The algorithm represents the use of the census transform operator over the temporal axis with an additional step of comparison with the temporal mean and sum of two elements. The transformation is done by encoding the temporal grayscale values of pixels via the following steps:Pairwise comparison of neighboring and non-neighboring brightness values. If value A is bigger than value B, encode this as 1, else 0.Comparison of brightness values with the temporal mean brightness values. If value A is bigger than mean M, encode this as 1, else 0.Comparison of sums of two brightness values with all other neighboring and non-neighboring sums of two brightness values. If the sum of two brightness values A is bigger than the sum of two brightness values B, encode this as 1, else 0.

The terms neighboring and non-neighboring refer to the temporal (axis t) and not the spatial distance (axis *x* and *y*). 

The main benefit of transforming grayscale values to a lower-dimensional encoding is the reduction in computational complexity. The end result of this census transform is a bitstring, which, in this case, was limited to a length of 64 bits. Because of this, simpler similarity measures can be used for feature comparison (e.g., Hamming distance). The total feature length in bits depends on the number of temporal brightness values. 

### 3.4. Disparity Calculation

Disparity calculation is accomplished, using a sliding window architecture. Sliding window buffers are well-known structures that are widely used for solving many signal processing problems on FPGA-s [9,10]. Usage of sliding window buffers enables the construction of deep pipeline structures. Therefore, a properly utilized sliding window approach can be used for the pipelined disparity calculation on rectified images.

A high-level overview of the pipelined disparity calculator is presented in Figure 4. 

The operation of such a design can be described in the following manner:
Two input features (F_R_ and F_L_), each encoded as a 64-bit long bitstring, are moved into the calculator during each operating cycle. The calculator includes a buffer, which represents the sliding window. The buffer consists of D memory cells. Each cell holds a feature (64-bit long bitstring) from the right camera view. Every memory cell can be accessed simultaneously during each clock cycle. When a new feature (in this case F_R_) is fed into the calculator, it enters the buffer at the head of the buffer (index D-1) and shifts all other features by one element to the left. The left-most element (index 0) is shifted out of the calculator. This process repeats every cycle.The second input feature, feature (F_L_), is read into the calculator and stored into a single memory cell. Since we are calculating the left-to-right disparity, we only require one memory cell, which holds the temporal feature from the left view (F_L_). This input feature is accessed by every processing element (PE) during the calculation in this cycle.The calculator includes D processing elements (PEs). Each of these processing elements accesses the cell with the F_L_ feature and the respective F_R_[i] feature cell in the sliding window, where *i* represents the PEs index. Each processing element calculates the Hamming distance, using inputs from the two respective memory cells, and saves the output to the respective H[i] cell of the Hamming distance buffer, H.The calculated Hamming distances in the H buffer represent the input into the minimum cost sorting network. Using a divide-and-conquer approach, neighboring elements of the buffer are pairwise compared. Through a series of comparisons, the minimum element and the respective index of the minimum element are determined. The resulting index of the element of the smallest Hamming distance represents the disparity. The expected disparity value of such a design ranges from -D_H_ to D_H_ [-D_H_, D_H_).

From an image-processing perspective, we are trying to find the closest match between a single 64-bit feature from the left image in a pool of D numbers of 64-bit feature candidates from the right image. When calculating the right-to-left disparity, the roles are reversed.

An additional prefill read procedure is required in order for this design to function as described above. When calculating the left-to-right disparity, a total of D/2 read cycles of the right input feature F_R_ are required to be read in order to complete the prefill procedure. The prefill procedure is necessary in order to fill the shift buffer and thus, establish the candidate pool for the disparity search. For example, if we are calculating the left-to-right disparity at image location x = 0, y = 0, we require access to the feature F_L_(0,0) and all features F_R_(x, 0), where x ranges from 0 to D/2 [0, D/2).

After the initial prefill, this design can calculate the disparity in every clock cycle if sufficient hardware resources are available. The HLS pseudocode of the above-described design is displayed in Figure 14.

**Listing 2 sensors-21-06435-f014:**
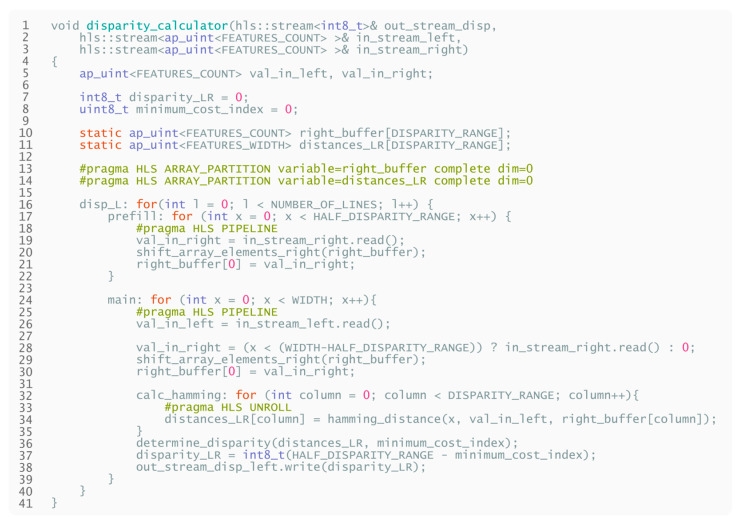
High-level pseudocode for disparity calculation. Both left-to-right disparity and right-to-left disparity can be calculated simultaneously if adequate resources are available.

### 3.5. Consistency Checker

Consistency checking is performed using the following steps:The previously calculated left-to-right disparity (D_LR_) and right-to-left disparity (D_RL_) are moved into the consistency checker during each operating cycle.A buffer, consisting of D memory cells, represents the sliding window. Every time the value of D_LR_ is read, it enters the buffer head (index D-1). All other elements get shifted by one element to the left. The element at the buffer tail is shifted out. This process repeats every cycle.The right-to-left disparity (D_RL_) is used as the input of the select port for the multiplexer. Disparity represents the index difference of corresponding temporal pixels from different views. This fact can be exploited when determining consistency. If we take the LR disparity D_LR_ and RL disparity D_RL_, we expect that the contents of the memory cell at the D/2 + D_RL_ index of the buffer D_LR_ will contain the value −D_RL_ = D_LR_. If this condition is fulfilled, the result is considered consistent. Since we are working with coarse disparities, we decided to use a less strict condition and considered the disparity D_LR_ to be consistent under the condition that |D_LR_ − D_RL_| ≤ 1 is satisfied.

The structure of the consistency checker is displayed in Figure 5.

The HLS pseudocode describing the above-presented behavior is displayed in Figure 15.

**Listing 3 sensors-21-06435-f015:**
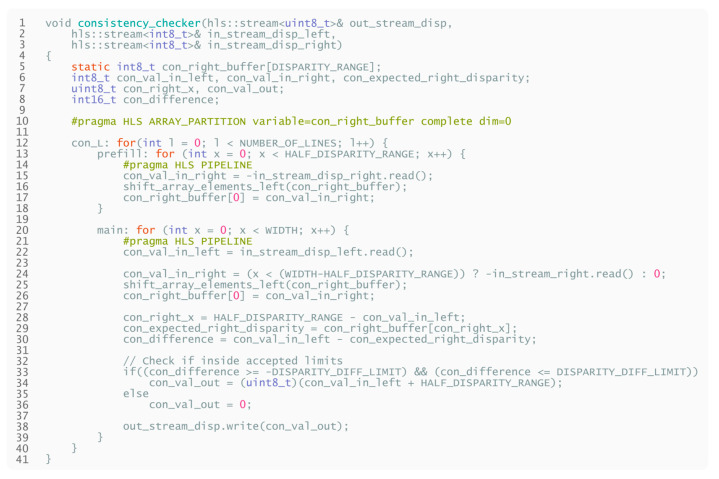
High-level pseudocode for consistency checking.

### 3.6. Median Filter

Median filtering represents one of the tasks that has already been evaluated extensively on many FPGA platforms [11,12,13]. Median filters can be easily realized using FPGA hardware and are commonly implemented using a sliding window buffer and compare-and-swap sorting network. The sorting network can be realized using a systolic array in order to enable deep pipelining.

The structure of the sliding window and the sorting network is displayed in Figure 6.

The approach displayed in Figure 6 requires two sliding windows. The first window is the median filter window, which holds the pixels that are input into the sorting network. The second window is the line buffer. This design only requires one read operation for every pixel and can, therefore, be easily pipelined. The values of the median filter window are obtained by accessing the local values stored in the line buffer.

The used sorting network can be described, using a series of compare-and-swap operations. By adding intermediate buffers after each compare-and-swap operation, this design can reach single clock cycle pipeline operation. The overall structure can be modified to be used with any image width and median filter window size.

### 3.7. Subpixel Refinement

Subpixel refinement represents the last step in the signal processing pipeline. Subpixel refinement makes it possible to obtain disparity maps that exceed the nominal pixel resolution of the input image. 

Refinement on a subpixel level is done by recalculating disparity on a smaller search range, the bounds of which are determined by the initial coarse correspondence search. The inputs for calculation are not bitstrings (temporal features obtained using the census transform), but temporal pixels in the form of grayscale values. In order to refine the initial estimate, we use the normalized cross-correlation similarity measure for calculating the similarity between different temporal pixels. The final subpixel value is determined, using parabolic interpolation. 

A high-level overview of the subpixel refinement design in displayed in Figure 7.

From Figure 7, it can be seen that the subpixel refinement process structure is similar to the structure used in disparity calculation. The main difference between these two designs is the underlying memory structure of the buffers. Both designs use a sliding window architecture, but the sliding window in the subpixel refinement design is realized using several parallel buffers (each buffer is implemented as a Dual-port Block RAM) instead of distributed memory (LUTRAM). 

Using Block RAM as the main memory structure reduces the total usage of LUTRAM, which is already used extensively during the initial coarse correspondence search. This balances the total resource utilization. 

The use of parallel Block RAM memory structures makes it possible to simultaneously access multiple temporal pixels within the refinement search range, as displayed in Figure 7. Here, the coarse disparity D_c_ is used for selecting the search range used for refinement. In Figure 7, the 125th column of the memory is selected as the refinement search range. The memory cell in 4th BRAM structure at the 125th column represents the center of the search range. 

The mode of access is scheduled to prevent simultaneous memory cell access when writing and reading. Scheduling enables simultaneous read/write operation without access collision. Writes that occur on the same clock cycle are displayed using the same color code. Greyed-out memory cells represent unused memory cells.

When looking at Figure 7, it can be seen that the temporal pixel T_R_ is written into Block RAM in diagonal writing order. For example, during the 127th clock cycle, TR is written to the 125th memory cell of the 7th BRAM structure, 126th cell of the 6th BRAM, 127th cell of the 5th BRAM, 128th cell of the 4th BRAM, 129th cell of the 3rd BRAM, 130th cell of the 2nd BRAM, 131th cell of the 1st BRAM and 132th cell of the 0th BRAM. A prefill procedure is required to prevent memory access collision in the same way as with every other sliding window approach used in other components.

The HLS pseudocode of the above-described design is displayed in Figure 16.

**Listing 4 sensors-21-06435-f016:**
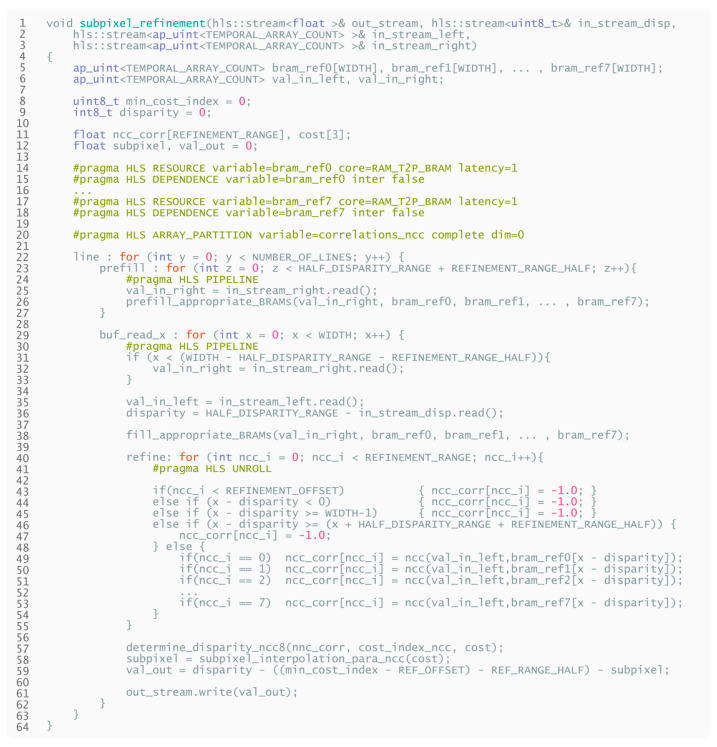
High-level pseudocode for subpixel refinement. Repetitive parts of the design are omitted to improve clarity.

## 4. Experiments and Results

### 4.1. Experimental Setup

We use two computing platforms in order to evaluate the performance of the above-described processing algorithms. The general experimental test setup overview is displayed in Figure 8.

For the first platform, we used the Xilinx ZC706 Evaluation Kit as the computing platform. This kit was used to evaluate the FPGA-based design.

The second platform was used to evaluate the GPU performance. The components of this platform are an Intel Core i7 7700, Nvidia GTX 1070. The GPU solution was not the main focus of this research but was used for comparative reasons in order to establish a best-effort reference implementation.

As previously mentioned, the dataset used for evaluation was taken from the initial publication explaining the extended census transform (referred to also as BICOS PLUS) [6]. The provided dataset includes 400 rectified image pairs of a scene illuminated by an aperiodic fringe pattern. The photographed test scene includes objects made from different surface materials. This includes a ceramic bust, a wooden chest, a ceramic cup with felt handle, a model of a polyhedral compound made from 3D-printed plastic, a disc and a turbine housing, both made from cast iron. The authors of the dataset note that an artificial light source in the form of an inhomogeneous NIR ambient light source was used to illuminate the scene from the left side in order to simulate real-world conditions.

As already noted by the authors in [6], the raw captured images of the test scene are subject to lens distortion and lens blur. The captured raw images are rectified, thus undistorting the image before being processed by the algorithm.

### 4.2. FPGA Hardware Utilization

Multiple different design configurations were synthesized in order to verify and evaluate the whole processing pipeline. All HLS designs were compiled, using Vivado HLS 2018.2. The obtained Verilog code was synthesized and implemented, using Vivado 2018.2.

The design parameters were set to the following values for evaluation:AXI4 Interface—64 bits data width, burst limit 256 transfers.Temporal census transform maximum feature count: 64.Image dimensions: 1024 × 1024 pixels, 10-bit grayscale pixels, rectified.Coarse disparity search range: 256 disparity levels [−128, 128).Median filter window size: 3 × 3 pixels.Subpixel disparity refinement search range: 8 disparity levels [−4, 4).Subpixel refinement interpolation: parabolic.

The above-mentioned parameters were kept constant during evaluation. The only parameter of the configuration that was changed was the number of successive temporal images used for evaluation (N).

The utilization report is displayed in Table 1 and Figure 9.

As seen in Figure 9, resource utilization follows a clear trend. The number of temporal images used for disparity calculation clearly increases the resource utilization of the proposed design. The amount of DSP and block RAM units used by the design rises linearly with N. The number of used flip-flops and LUTs rises until the number of features calculated by the temporal census transform reaches the preset maximum limit (64). Overall, the resource utilization of every resource is below 50%.

The resource utilization of individual building blocks for the configuration where N equals 10 is displayed in Table 2 and Figure 10.

The main contributors to the resource utilization are the disparity calculator and subpixel refinement component. The reason for such a high requirement of FF and LUT primitives by the disparity calculator is the large number of processing elements used for calculating the Hamming distance. When calculating the disparity range of 256 disparities per clock, we require a constant operation of 256 PEs to fulfill this requirement. Because we are calculating both LR and RL disparities in parallel, the required amount of PEs doubles to 512.

The overall greatest number of DSP primitives is required by the subpixel refinement component. Since refinement is done using normalized cross-correlation, an operation that requires extensive use of multiplication and division, the DSP usage increases drastically.

Block RAM usage is distributed over several components. The AXI read interface component uses this primitive for the buffering of input data, and the subpixel refinement component uses Block RAM for implementation of the sliding window buffer. The overall utilization of this primitive is still low.

### 4.3. FPGA Performance and Accuracy Results

The FPGA platform was used and tested as a standalone solution. A system clock frequency of 90.9 MHz was used for evaluation.

The power draw of the evaluation platform was measured, using a programmable DC power supply Tenma Model 72-2540. The measured power consumption indicates the power used by the entire evaluation kit.

The validity of every single subpixel disparity is determined separately. A subpixel disparity pixel is deemed valid if the difference between the ground-truth (disparity map calculated using 400 temporal samples) and subpixel disparity pixel is less than 1. If this threshold is exceeded, the result is deemed invalid. The ratio of valid disparity pixels and the sum of the count of valid pixels and count of invalid pixels represents the percentage of valid results. A visual example of the importance of validity is displayed in Figure 11.

As seen in Figure 11, the ground-truth disparity map displays accurate and valid depth information throughout the entire scene, with the exception of the reflective toggle latch located on the wooden chest. This indicates that the aperiodic fringe projection method using visible or NIR light is still subject to limitations, such as scanning highly transparent or reflective materials.

The performance results and power measurements are displayed in Table 3.

The values from Table 3 indicate that the maximum achieved output framerate is the highest for N = 5. A higher number of temporal images increases the quality (validity) of the results. At N = 8, we start to experience only diminishing effects when increasing the number of temporal samples. Power consumption is mostly irrelevant based on the number of temporal samples. The performance-per-watt expectedly decreases with a higher number of temporal samples, as the memory bandwidth cannot increase further without an increase in the AXI interface clock speed.

### 4.4. GPU Performance Results

During the evaluation of the GPU performance, the immediate power draw was determined by using the nvidia-smi software. The average (mean) power draw was then determined by averaging out the measured intermediate power values. The performance results are displayed in Table 4.

### 4.5. Comparison of Performance Results

The same performance tests were conducted on both computing platforms. The results of the tests can be summarized in a comparison. A general overview of the performance results obtained by both platforms is displayed in Figure 12.

The results indicate that the framerates achieved using the GPU platform are much higher than the framerates calculated by the FPGA platform. The power consumption scales together with performance. However, when looking at the energy efficiency of calculations, the FPGA platform achieves two-times higher performance-per-watt on average.

The work published by Dietrich et al. [6] only performed one performance measurement (64-bit features, N = 10, and other parameters are unknown). The achieved GPU performance on a GTX1080 was 52.6 FPS, but the measurement included the additional step of image rectification.

## 5. Discussion and Future Work

This paper has proposed a flexible HLS-based FPGA stereo-matching architecture for temporal statistical pattern projector systems. Furthermore, we provided an implementation capable of leveraging several FPGA-specific optimization primitives, such as sliding windows approaches, custom systolic arrays, and optimizations, such as instruction and task pipelining. The proposed design was tested on an off-the-shelf FPGA platform and achieved real-time disparity calculation results.

When evaluating the normalized performance-per-watt, the FPGA design achieved two-times better results than a highly optimized GPU solution. Further optimizations are possible, as no HDL-level optimizations were made during this work. The proposed work also did not fully utilize all available FPGA resources, so additional spatial parallelization can still be used to achieve higher disparity calculation framerates.

## Figures and Tables

**Figure 1 sensors-21-06435-f001:**
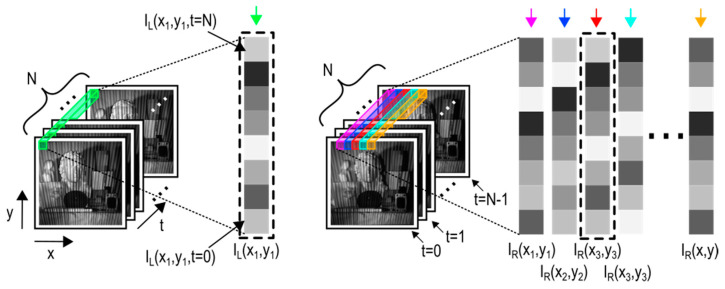
Correspondence search between individual “temporal pixels” from left (image stack I_L_) and right (image stack I_R_) view. The closest match to temporal pixel I_L_(x_1_, y_1_) (green arrow) is the temporal pixel I_R_(x_3_, y_3_) (red arrow).

**Figure 2 sensors-21-06435-f002:**
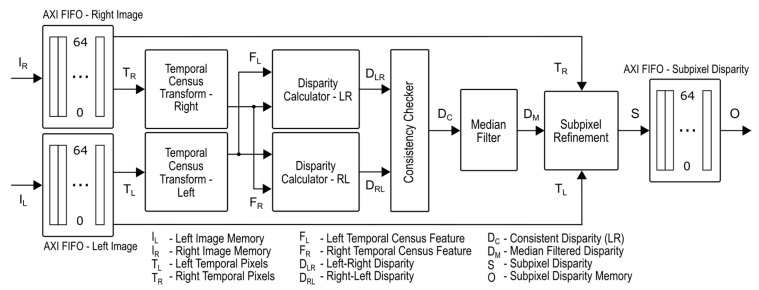
High-level structure of the proposed FPGA-based design.

**Figure 3 sensors-21-06435-f003:**
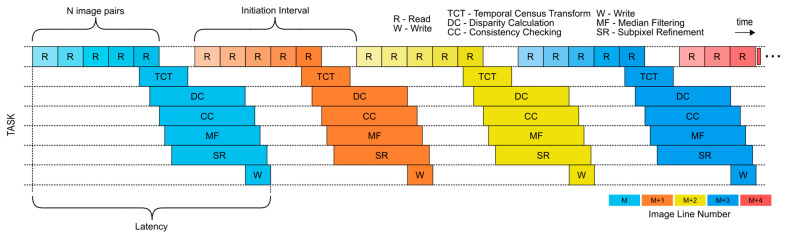
A high-level overview of task pipelining in the design.

**Figure 4 sensors-21-06435-f004:**
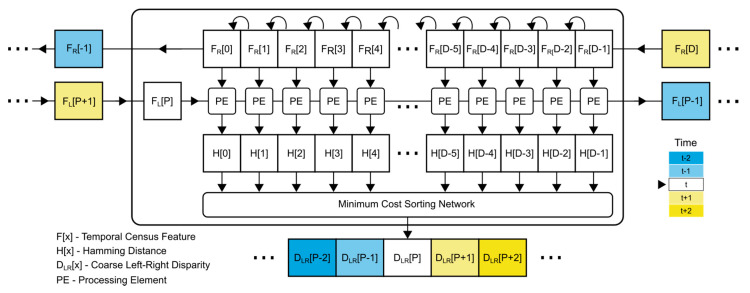
High-level overview of disparity calculation. The flow of data inside and into the design is displayed using arrows. Clock cycles are encoded, using different colors.

**Figure 5 sensors-21-06435-f005:**
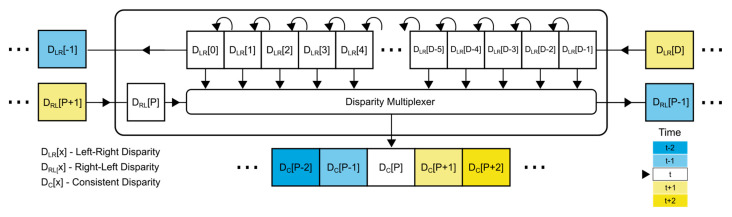
A high-level overview of the consistency checker design.

**Figure 6 sensors-21-06435-f006:**
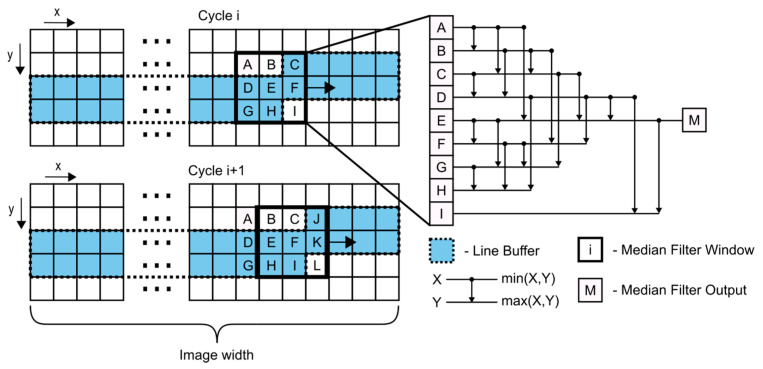
Overview of the median filter design. The left part displays the sliding window concept when filtering. The right part displays the sorting network. A 3 × 3 median window requires 19 compare-and-swap operations. CAS symbols are taken from [13].

**Figure 7 sensors-21-06435-f007:**
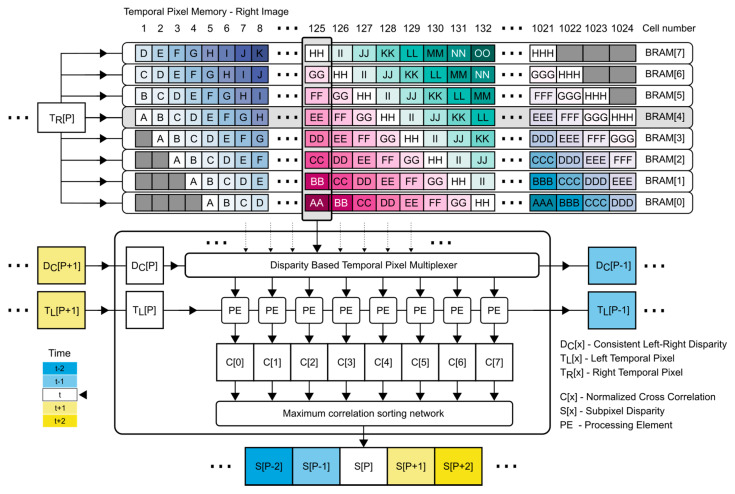
A high-level overview of the subpixel refinement process. Each Block RAM can hold a full image line of temporal pixels. Each read cycle, the right temporal pixel is saved into every Block RAM structure (8 in this case) at the respective memory locations (diagonal). Color codes indicate the clock cycle of writing. Characters are used to display the memory contents.

**Figure 8 sensors-21-06435-f008:**
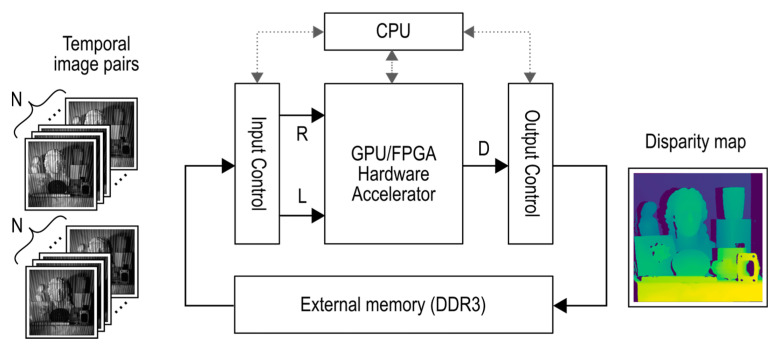
Experimental test setup using GPU or FPGA as the computing platform.

**Figure 9 sensors-21-06435-f009:**
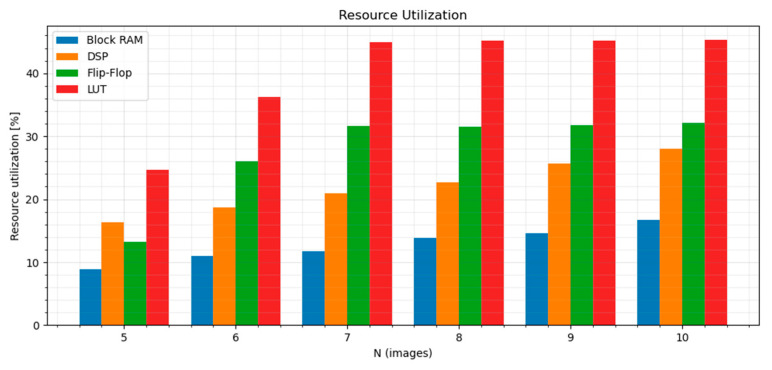
Resource utilization of individual FPGA primitives for the proposed design.

**Figure 10 sensors-21-06435-f010:**
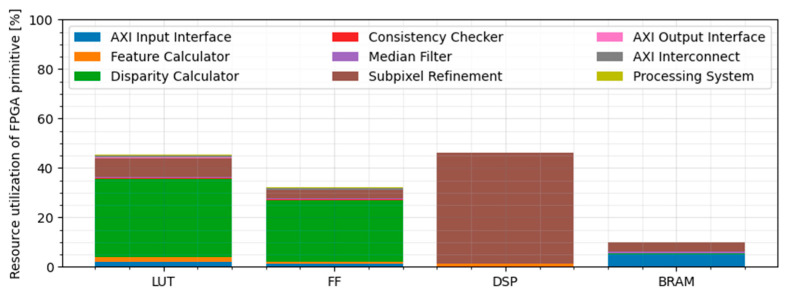
Resource utilization of individual FPGA primitives for the configuration N = 10.

**Figure 11 sensors-21-06435-f011:**
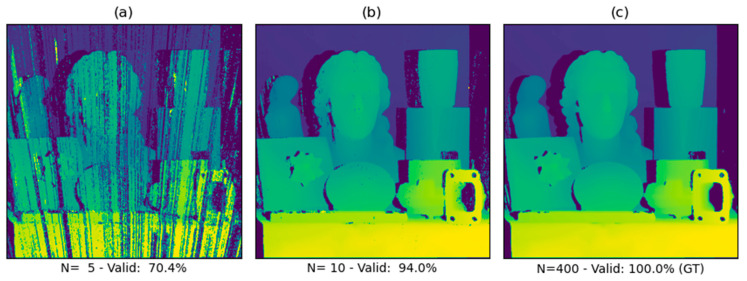
Overview of the validity of disparity results. (**a**) At N = 5, heavy artifacts are present. (**b**) At N = 10, only minor artifacts are present, mostly limited to the occluded areas of the disparity map. (**c**) At N = 400, there are almost zero or no artifacts. The last example is considered the ground truth for determining the validity.

**Figure 12 sensors-21-06435-f012:**
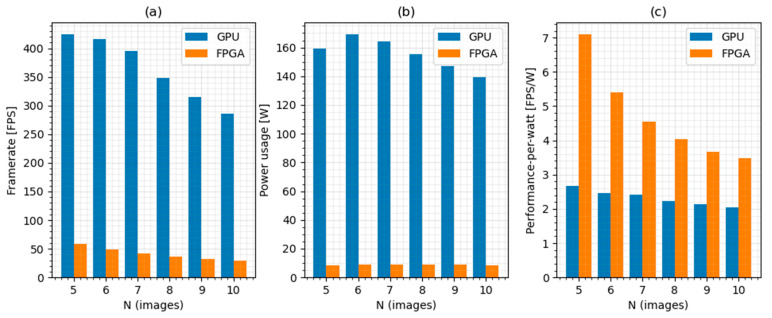
Overview of performance results. (**a**) Average measured framerate for GPU and FPGA. (**b**) Power usage for individual platforms. (**c**) Performance-per-watt for both platforms.

**Table 1 sensors-21-06435-t001:** Utilization report of the implemented design for individual FPGA primitives. The target FPGA device is XC7Z045FFG900.

		LUT		Flip-Flop		Block RAM		DSP	
N	f_max_ [MHz]	Total: 218,600	[%]	Total: 437,200	[%]	Total: 1090	[%]	Total: 900	[%]
5	96.04	54,102	24.7	57,505	13.2	97	8.9	147	16.3
6	93.63	79,174	36.2	114,203	26.1	120	11.0	168	18.7
7	92.6	98,172	44.9	138,090	31.6	128	11.7	189	21.0
8	91.1	98,788	45.2	137,773	31.5	150	13.9	204	22.7
9	92.2	98,762	45.2	139,111	31.8	159	14.6	231	25.7
10	93.6	99,019	45.3	140,388	32.1	182	16.7	252	28.0

**Table 2 sensors-21-06435-t002:** Utilization report of the implemented design at N = 10. * The AXI Interconnect component is a Xilinx IP block. The processing system (ARM Cortex PS) is implemented in silicon but requires some logic utilization for interfacing.

	LUT		Flip-Flop		Block RAM		DSP	
Component	Total: 218,600	[%]	Total: 437,200	[%]	Total: 1090	[%]	Total: 900	[%]
AXI Read Interface	4201	1.9	5096	1.2	45	8.3	0	0.0
Feature Calculator	4051	1.9	3063	0.7	0	0.0	6	0.7
Disparity Calculator	69,355	31.7	109,365	25.0	4	0.7	0	0.0
Consistency Checker	688	0.3	2167	0.5	0	0.0	0	0.0
Median Filter	857	0.4	416	0.1	5	0.9	0	0.0
Subpixel Refinement	16,896	7.7	15,499	3.5	36	6.6	246	27.3
AXI Write Interface	393	0.2	777	0.2	1	0.2	0	0.0
AXI Interconnect *	2087	1.0	3329	0.8	0	0.0	0	0.0
Processing System *	491	0.2	646	0.1	0	0.0	0	0.0

**Table 3 sensors-21-06435-t003:** Performance report of the FPGA solution.

N	Input Framerate [FPS]	Output Framerate [FPS]	Valid [%]	Power Consumption [W]	Performance-Per-Watt [FPS/W]
5	291	58.2	70.4	8.2	7.10
6	292.2	48.7	88.2	9.0	5.41
7	293.3	41.9	92.0	9.2	4.55
8	294.4	36.8	93.0	9.1	4.04
9	294.3	32.7	93.6	8.9	3.67
10	295.0	29.5	94.0	8.5	3.47

**Table 4 sensors-21-06435-t004:** Performance report of the GPU solution.

N	Output Framerate [FPS]	Valid [%]	Power Consumption [W]	Performance-Per-Watt [FPS/W]
5	424.6	70.4	159.2	2.67
6	416.4	88.2	169.2	2.46
7	396	92.0	164.4	2.41
8	348	93.0	155.5	2.24
9	314.5	93.6	147	2.14
10	285.8	94.0	139.2	2.05

## Data Availability

Not applicable.

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
