# Peer review of "Real-Time FPGA Accelerated Stereo Matching for Temporal Statistical Pattern Projector Systems"

_sensors, 2021, doi:10.3390/s21196435_

Round 1

Reviewer 1 Report

This paper proposed a flexible HLS based FPGA stereo matching architecture for temporal statistical pattern projector systems. The provided an implementation capable of leveraging several FPGA-specific optimization primitives such as sliding windows approaches, custom systolic arrays, and optimization such as instruction and task pipeline. 

And it is clear to read and the techniques proposed in the paper are also very useful for the other engineers in the same domain.

In the paper, the proposed work did not fully utilize all available FPGA resources, and I think the spatial parallelization is important for the calculation frame rates and the performance. And in the design architecture, the authors should consider the spatial parallelization and its expansion.

Reviewer 2 Report

This paper focuses on the algorithmic aspects related to the optical depth measurement in the case of the use of an active system using a pattern projected on the scene to be evaluated. The implemented technique is based on the projection of an aperiodic sinusoidal fringe pattern. The authors propose an FPGA architecture in order to solve the problem of the computational load (and thus of the implemented devices) while ensuring a portability of the system taking into account both the importance of the dedicated electronics and its energy consumption.

The paper is structured in five sections. After a general introduction, the authors provide a state of the art and give the general context of their study. Then, they present the detailed architecture of their device through an extensive study of the structure in different blocks, algorithms presented in pseudo-code and the different optimizations they have performed.Then, the authors present the experimental results obtained on a published dataset, thanks to an implementation of their system from the point of view of both accuracy and computational performance.  Finally, the authors bring a complete conclusion through a discussion of their results and the future works envisaged.

The paper is globally clear, well written and well structured. It is based on a complete and serious bibliography. The results presented are convincing and confirm the expectations. The conclusions and perspectives presented are consistent with the elements presented in the previous sections.

If I may make a remark, I regret that the experimental device of shooting and the characteristics of the scene are not questioned. Indeed, in this type of application, the imaging conditions can be of great importance and the nature of the scene, its illumination as well as the types of cameras used (in particular their optics) can impact the result in a significant way. I fully understand that an exhaustive study of the different imaging conditions is not within the scope of this paper and that the use of a published data set seems sufficient. However, it would be desirable to question the validity of the approach and to mention its limitations (in a few lines).

Nevertheless, I do not consider that this constitutes an objection to the publication as it is, but that it would constitute an improvement of a paper that is already of excellent level.
